# The Neuroprotective Effects of Peripheral Nerve Microcurrent Stimulation Therapy in a Rat Model of Middle Cerebral Artery Occlusion

**DOI:** 10.3390/ijms251810034

**Published:** 2024-09-18

**Authors:** Yoon-Jin Lee, Eun Sang Kwon, Yong Suk Moon, Jeong-Rang Jo, Dong Rak Kwon

**Affiliations:** 1Department of Biochemistry, College of Medicine, Soonchunhyang University, Cheonan 31151, Republic of Korea; leeyj@sch.ac.kr; 2Department of Medicine, College of Medicine, Keimyung University, Daegu 42601, Republic of Korea; peterkwon031218@gmail.com; 3Department of Anatomy, Catholic University of Daegu School of Medicine, Daegu 42472, Republic of Korea; ysmoon@cu.ac.kr; 4Department of Rehabilitation Medicine, Catholic University of Daegu School of Medicine, Daegu 42472, Republic of Korea; kbbio123888@gmail.com

**Keywords:** microcurrent therapy, middle cerebral artery occlusion, neuroprotection, peripheral nerve stimulation, stroke recovery

## Abstract

This study investigated the neuroprotective effects of peripheral nerve microcurrent stimulation therapy in a rat model of middle cerebral artery occlusion (MCAO). Twenty 8-week-old male Sprague Dawley rats weighing 300–330 g were categorised into group A, serving as the healthy control; group B, including rats subjected to MCAO; group C, including rats receiving microcurrent therapy immediately after MCAO, which was continued for one week; and group D, including rats receiving microcurrent therapy one week before and one week after MCAO. A gross morphological analysis, behavioural motion analysis, histological examination, immunohistochemistry, and Western blotting were conducted. Microcurrent therapy significantly reduced ischaemic damage and pyramidal cells of the hippocampus CA1 region. Haematoxylin and eosin staining revealed infarction areas/viable pyramidal cell numbers of 0%/94.33, 28.53%/40.05, 17.32%/80.13, and 5.38%/91.34 in groups A, B, C, and D, respectively (*p* < 0.001). A behavioural analysis revealed that the total distances moved were 1945.24 cm, 767.85 cm, 1781.77 cm, and 2122.22 cm in groups A, B, C, and D, respectively (*p* < 0.05), and the mean speeds were 6.48 cm/s, 2.50 cm/s, 5.43 cm/s, and 6.82 cm/s, respectively (*p* < 0.05). Inflammatory markers (cluster of differentiation 68, interleukin-6, and tumour necrosis factor-α) significantly decreased in the treated groups (*p* < 0.001). Western blotting revealed reduced proinflammatory, oxidative stress, and apoptosis-related protein levels, along with increased angiogenic factors and mitogen-activated protein kinase (MAPK) pathway modulation in the treated groups. Peripheral nerve microcurrent stimulation therapy effectively mitigates ischaemic damage, promotes recovery, reduces inflammation, and modulates protein expression, emphasising its potential as a therapeutic strategy for ischaemic stroke.

## 1. Introduction

Globally, stroke represents a critical health issue, causing significant morbidity and mortality. Demographic ageing and unhealthy lifestyle choices among younger populations increase the prevalence of stroke, causing profound personal suffering and substantial socio-economic repercussions [1,2]. Ischaemic strokes, caused by blood vessel blockages, account for >80% of cases, with the middle cerebral artery being the most affected [3].

Thrombolysis with recombinant tissue plasminogen activator (tPA) is currently the only approved acute treatment for ischaemic stroke. However, a narrow therapeutic window, modest reperfusion success rates, risks of haemorrhagic transformation, and potential neurotoxic effects limit its use [4]. These limitations warrant innovative therapeutic strategies to improve outcomes for patients with stroke.

Sudden blood vessel blockages that interrupt brain blood flow, resulting in cellular malfunctions, cause ischaemic strokes. The lack of oxygen and nutrients disrupts mitochondrial activity and ATP production, causing reactive oxygen species (ROS) overproduction, which is harmful to cellular structures [5]. ATP scarcity undermines ion transporters, causing calcium ion accumulation and thereby exacerbating cell damage [6,7]. Additionally, excitotoxicity caused by glutamic acid and heightened oxidative stress contributes to brain tissue swelling, inflammatory pathway activation, and cell death [7,8].

The mechanisms through which microcurrent therapy exerts its effects are not fully elucidated, but several hypotheses are supported by existing research. Microcurrent therapy is believed to enhance cellular energy production, improve amino acid transport, and stimulate protein synthesis, which can collectively reduce inflammation and support tissue repair [9]. Persistent neuroinflammation is a significant factor in neurodegenerative diseases and cerebrovascular events, such as stroke [10].

When applied peripherally, such as to the femoral nerve, microcurrent therapy may initiate systemic responses through both humoral and neural pathways. For instance, stimulating the femoral nerve could lead to the release of blood-borne factors, including small molecules and neurotrophic factors that travel to the central nervous system [11]. Additionally, neural pathways might be activated, potentially indirectly modulating central nervous system processes [12]. While the exact pathways remain under investigation, current evidence suggests that these mechanisms could contribute to the neuroprotection and modulation of neuroinflammation in stroke models.

Our previous studies indicate that applying microcurrent electrical stimulation therapy to peripheral nerves may mitigate cognitive decline and neuroinflammation in Alzheimer’s disease [13,14]. This therapy utilises a very low electric current, typically < 1000 µA, which is imperceptible to the patient. It modulates neuroinflammation via the mitogen-activated protein kinase (MAPK) and Toll-Like Receptor 4 (TLR4) signalling pathways, potentially reducing neuroinflammatory proteins and improving ischaemic stroke outcomes.

This study investigates the effects of microcurrent therapy on improving motor functions and activity levels, which enhance recovery post-stroke, considering the need for noninvasive therapies that effectively model ischaemic stroke without causing further harm. We hypothesise that microcurrent therapy will not only reduce the infarction area and neuronal loss but also modulate the activity of glial cells, such as astrocytes and microglia, which are critical mediators of inflammation in the brain. By targeting these cells, microcurrent therapy may decrease proinflammatory cytokine production, reduce apoptosis, and promote neurovascular repair through increased VEGF expression. This multifaceted approach is expected to enhance overall recovery in ischaemic stroke. 

We aim to elucidate how microcurrent therapy modulates the key proteins involved in inflammation, angiogenesis, and apoptosis, providing insights into its neuroprotective effects. This study could pave the way for clinical applications, providing a noninvasive, effective, and safe therapeutic strategy for stroke rehabilitation.

## 2. Results

### 2.1. Gross Morphological Comparison of Brain Tissue among Groups

A gross morphological comparison across these groups emphasises the effectiveness of microcurrent therapy in mitigating the effects of MCAO. The control group maintains a normal brain structure, whereas the disease group demonstrates significant damage and necrosis. The post-treatment images exhibit notable recovery, indicating that microcurrent therapy contributes to structural restoration and ischaemic damage reduction (Figure 1).

### 2.2. Motion Analysis

The behavioural motion analysis depicted in Panels A and B indicates significant differences among the experimental groups. Group A recorded an average of 1945.24 cm (±88.96) in terms of the total distance moved (Panel A), which is significantly higher than group B, which only moved 767.85 cm (±57.58, *p* < 0.05). Group C increased the total distance to 1781.77 cm (±446.37), demonstrating notable recovery, whereas group D achieved the highest recovery with an average of 2122.22 cm (±498.33), almost reaching control levels.

Similarly, the mean speed in the zone (Panel B) followed a comparable trend. Group A maintained the highest mean speed at 6.48 cm/s (±0.30), which is significantly faster than group B’s 2.50 cm/s (±0.42, *p* < 0.05). Group C demonstrated improved speeds of up to 5.43 cm/s (±0.92), whereas group D further increased to 6.82 cm/s (±1.95), nearly matching group A’s performance.

### 2.3. Histological Infarction Area Analysis

High magnification of the H&E-stained slides demonstrated the infarction area of the brain tissue and its boundaries. A histological examination revealed no areas with brain tissue damage in the ipsilateral or contralateral hemisphere in the group A rats. Extensive tissue oedema and necrosis were observed in the cortex, striatum, and thalamus of the left cerebral hemisphere in group B, and damaged neurons were observed in the hippocampus. The infarct area was reduced in the cortex and striatum in groups C and D.

H&E staining and the subsequent quantification of infarction areas revealed significant differences among the groups. The total infarction area was notably higher at 28.53 ± 1.48% in group B compared to group A, exhibiting no infarction (0%). Group C significantly reduced the infarction area to approximately 17.32 ± 1.16%, whereas group D demonstrated an even more substantial reduction, with the infarction area being close to 5.38 ± 0.71% (Figure 2).

The infarction area in group B was 30.03% ± 2.50% at the bregma +1 ± 0.5 mm level, whereas it was significantly lower in group C at 23.17% ± 1.95% and group D at 10.11% ± 1.72%. Similarly, group B demonstrated the highest infarction area at 41.20% ± 1.87% at the bregma −2 ± 0.5 mm level compared to 23.49% ± 1.68% in group C and 4.95% ± 0.74% in group D (Figure 2). Finally, the infarction area was 14.35% ± 0.74% in group B at the bregma −5 ± 0.5 mm level, reduced to 5.29% ± 0.53% in group C, and further decreased to 1.08% ± 0.18% in group D (Figure 2).

### 2.4. Histopathology in the Hippocampus

Figure 3 shows representative images of the hippocampus CA1 regions. Viable pyramidal cells displayed normal histological features and were arranged in regular orientation in group A. In contrast, degenerated pyramidal cells in group B demonstrated irregular neuronal arrangements and darkly stained and vacuolated nuclei. Compared with group B, the pyramidal cells appeared more ordered and darkly stained, and vacuolated nuclei were reduced in groups C and D. H&E and CV staining and the subsequent quantification of infarction areas revealed significant differences among the groups. The total numbers of viable pyramidal cells in a 1 mm range of the CA1 region were notably lower at 40.05 ± 2.79 in group B compared to group A, which exhibited no pathology (94.33 ± 0.36). Group C significantly increased the pyramidal cell numbers to 80.13 ± 1.34, whereas group D demonstrated an even more substantial improvement, with the pyramidal cell numbers being close to 91.34 ± 0.40. The percentage of degenerative pyramidal cells in a 1 mm range of the CA1 region in groups A, B, C, and D were 5.66% ± 0.36%, 59.94% ± 2.79%, 19.86% ± 1.34%, and 8.65% ± 0.40%, respectively. The proportion of degenerated pyramidal cells was significantly decreased in groups C and D compared to group B.

### 2.5. The Anti-Inflammatory Effects of Microcurrent Therapy in Infarction Areas: Immunohistochemical and Western Blot Analyses

We conducted immunohistochemical staining for CD68, IL-6, and TNF-α to assess inflammation. The analysis showed no significant staining for these markers in group A, indicating an absence of inflammation in this group. Group B showed intense staining across the total area, with a high positive area of CD68 (30.41% ± 1.38%). Group C (16.84% ± 0.94%) showed reduced staining and a positive area compared to group B, and group D (7.87% ± 0.98%) exhibited minimal staining with positive areas nearing those of group A. The IL-6 staining results are similar, with no significant positive area being found in group A (0%), intense staining in group B (31.15% ± 1.89%), reduced staining in group C (14.29% ± 0.98%), and minimal staining in group D (4.68% ± 0.61%). For TNF-α staining, group A showed no significant positive area (0%), group B had intense staining (31.46% ± 1.94%), group C had reduced staining (15.12% ± 0.94%), and group D had minimal staining (5.81% ± 0.73%) (Figure 4A,B, Figure 5A,B and Figure 6A,B).

The positive areas for CD68, IL-6, and TNF-α in group B were 29.18% ± 1.91%, 31.76% ± 3.11%, and 31.82% ± 3.34% at the bregma +1 ± 0.5 mm level, whereas they were significantly lower in group C at 20.05% ± 1.63%, 20.76% ± 1.55%, and 20.60% ± 1.48% and in group D at 12.48% ± 2.22%, 7.30% ± 1.40%, and 8.53% ± 1.47% (Figure 4A,B, Figure 5A,B and Figure 6A,B). Similarly, group B demonstrated the highest positive areas at 43.69% ± 1.92%, 46.14% ± 2.92%, and 46.97% ± 2.96% at the bregma -2 ± 0.5 mm level compared to 22.61% ± 1.32%, 18.28% ± 1.42%, and 19.97% ± 1.29% in group C and 9.63% ± 1.47%, 6.14% ± 0.90%, and 8.53% ± 1.25% in group D (Figure 4A,B, Figure 5A,B and Figure 6A,B). Finally, the positive areas were 18.37% ± 1.32%, 15.55% ± 1.61%, and 15.60% ± 1.48% in group B at the bregma -5 ± 0.5 mm level, which reduced to 7.86% ± 0.77%, 3.82% ± 0.31%, and 4.79% ± 0.39% in group C and further decreased to 1.49% ± 0.36%, 0.60% ± 0.06%, and 0.37% ± 0.05% in group D (Figure 4A,B, Figure 5A,B and Figure 6A,B).

Immunopositivity for CD68, IL-6, and TNF-α increased microglia, macrophages, lymphocytes, and monocytes in the infarction area. Group A served as the baseline control, with no inflammatory markers observed (0%). In contrast, group B showed elevated levels of inflammatory markers with CD68-positive cells (5620.00 ± 529.75), IL-6-positive cells (1421.25 ± 129.90), and TNF-α-positive cells (1900.00 ± 110.73), as depicted in Figure 4C, Figure 5C and Figure 6C.

Group C demonstrated a significant reduction in inflammatory markers compared to group B, with CD68-positive cells (2462.50 ± 191.46), IL-6-positive cells (701.25 ± 62.89), and TNF-α-positive cells (993.75 ± 58.51), as shown in Figure 4C, Figure 5C and Figure 6C. Group D exhibited a further reduction in inflammatory markers, approaching baseline levels with CD68-positive cells (968.75 ± 218.30), IL-6-positive cells (701.25 ± 62.89), and TNF-α-positive cells (555.00 ± 60.16), as shown in Figure 4C, Figure 5C and Figure 6C.

The Western blot images provide a quantitative measure of the protein levels corresponding to the inflammatory markers CD68, IL-6, and TNF-α across the different groups, corroborating the immunohistochemical findings. Group A exhibited baseline levels of these inflammatory proteins: CD68 (1.00 ± 0.08), IL-6 (1.00 ± 0.05), and TNF-α (1.00 ± 0.11). Conversely, group B demonstrated significantly elevated levels: CD68 (3.64 ± 0.09), IL-6 (3.22 ± 0.17), and TNF-α (3.39 ± 0.07). Group C showed reduced levels: CD68 (2.70 ± 0.07), IL-6 (2.10 ± 0.19), and TNF-α (2.50 ± 0.43). Group D exhibited further reductions, approaching baseline levels: CD68 (0.83 ± 0.12), IL-6 (0.87 ± 0.07), and TNF-α (0.70 ± 0.08) (Figure 4D, Figure 5D and Figure 6D).

The accompanying bar charts quantify the differences in the inflammatory markers among the groups, demonstrating significant reductions in groups C and D compared to group B, with group D showing the most pronounced improvement.

Microcurrent therapy has demonstrated substantial efficacy in mitigating inflammation in the infarction area, progressively reducing the levels of inflammatory markers from group B to group D, approaching the baseline levels observed in group A.

### 2.6. Western Blot Analysis

#### 2.6.1. The Effect on the Expression of Immune-Related Proteins

The MMP8, NF-kB, and IL-1β expression levels were compared using Western blotting to determine the effect of microcurrent treatment on proinflammatory protein expressions. The MMP8 levels were 1.00 ± 0.70, 3.73 ± 0.43, 2.50 ± 0.14, and 0.86 ± 0.11, those for NF-kB were 1.00 ± 0.02, 4.08 ± 0.52, 1.84 ± 0.32, and 1.26 ± 0.18, and those for IL-1β were 1.00 ± 0.05, 3.63 ± 0.49, 2.00 ± 0.13, and 1.05 ± 0.07 in groups A, B, C, and D, respectively. The expression level of inflammation-related proteins significantly increased in group B compared to group A, but it was reduced by the microcurrent treatment, and group D demonstrated no significant difference compared to group A (Figure 7A).

The expression levels of TLR4 and MyD88, TLR-related markers, were compared. The TLR4 expression levels were 1.00 ± 0.26, 3.16 ± 0.16, 1.78 ± 0.09, and 1.24 ± 0.03 and those of MyD88 were 1.00 ± 0.15, 3.13 ± 0.24, 1.52 ± 0.05, and 1.08 ± 0.06 in groups A, B, C, and D, respectively, but the microcurrent treatment reduced the expression levels in group B (Figure 7A). 

#### 2.6.2. Effect on Angiogenic Factor Expression

The expression changes in VEGF, the most central factor in angiogenesis, were measured using Western blotting. VEGF expression was significantly decreased in group B (0.22 ± 0.04) compared to group A (1.00 ± 0.08), but it was significantly increased by the microcurrent treatment (0.85 ± 0.25 and 0.90 ± 0.17, respectively) (Figure 7B).

#### 2.6.3. Effect on Apoptosis-Related Expression

Changes in the expression levels of apoptosis-inducing factors caspase-3 and PARP, apoptosis-inducing protein BAX, and apoptosis-inhibiting protein Bcl-2 were measured using Western blotting to investigate the effect of microcurrent on apoptosis. Caspase-3 activation, which appears when cell death progresses, was significantly increased in group B (3.18 ± 0.15) compared to group A (1.00 ± 0.12) and was significantly reduced in groups C (1.19 ± 0.05) and D (1.00 ± 0.06). In addition, cleaved-PARP expression due to caspase-3 activation was significantly increased in group B (3.27 ± 0.93), but its expression was significantly reduced in groups C (1.47 ± 0.08) and D (1.07 ± 0.06), which underwent the microcurrent treatment. Apoptosis-induced protein BAX was significantly increased in group B compared to group A (group A, 1.00 ± 0.05; group B, 3.16 ± 0.19; group C, 1.25 ± 0.06; and group D, 1.01 ± 0.04) but decreased in the groups that underwent microcurrent treatment (groups C and D). Conversely, the apoptosis inhibitory protein Bcl-2 expression level was the lowest in group B, with expression levels of 1.00 ± 0.08, 0.45 ± 0.04, 1.03 ± 0.04, and 1.02 ± 0.03 in groups A, B, C, and D, respectively (Figure 8A).

#### 2.6.4. Effect on MAPK Protein Expression

Changes in ERK and p38 were evaluated through Western blotting to investigate the effect of microcurrent on MAPK activity. Group B demonstrated significantly increased phosphorylation in ERK and p38 compared to group A, and the microcurrent treatment was confirmed to reduce the protein expressions of ERK and p38. In particular, protein expression in group D was suppressed at a significantly higher level than in group A (Figure 8B).

#### 2.6.5. The Effect on DNA-Damage-Related Expression

Changes in p-Chk1 and p-Chk2 were evaluated through Western blotting to investigate the effect of microcurrent on DNA damage. Group B significantly increased phosphorylation in Chk1 and Chk2 compared to group A, and no significant difference in p-Chk1 was found between groups B and C, but p-Chk2 demonstrated a decrease in protein expression during the microcurrent treatment. In particular, protein expression in group D was suppressed similarly to group A (Figure 8C). 

## 3. Discussion

This study revealed that microcurrent therapy significantly mitigates inflammation, oxidative stress, apoptosis, and angiogenesis-related changes induced by MCAO in rats. It also improves behavioural recovery by increasing movement distance and speed. Microcurrent therapy also effectively reduces inflammatory, oxidative stress, and apoptotic marker expression, improves angiogenic factor levels, and normalises MAPK activity, indicating its potential therapeutic benefits in stroke recovery.

Microcurrent therapy effectively ameliorates MCAO-induced ischaemic stroke effects, as exhibited by gross morphological and histological analyses. The control group, demonstrating a normal brain structure, contrasts sharply with the disease group exhibiting necrosis and neuronal damage. The post-treatment observations reveal substantial brain structure recovery, indicating microcurrent therapy’s role in both structural restoration and reducing ischaemic damage. The treated groups demonstrated notable reductions in infarction areas across various bregma levels, emphasising microcurrent therapy’s ability to mitigate tissue damage and underscoring its efficacy [15,16,17,18].

A comparative analysis with previous studies supports microcurrent therapy’s neuroprotective benefits [13,14]. In this study, it effectively preserved neuronal integrity and reduced infarction size in an ischaemic stroke model. It improved histopathological outcomes in the treated groups, resulting in improved neuronal organisation and reduced cellular damage markers in crucial brain regions such as the hippocampal CA1 area, further underscoring its potential. These results collectively support microcurrent therapy as a promising intervention for ischaemic stroke, requiring further clinical exploration and application. 

We specifically selected the hippocampal CA1 region for our analysis due to its heightened vulnerability to ischaemic and oxidative damage. Brain tissue, including the CA1 region, is known to be particularly susceptible to oxidative stress [19], and several studies have indicated that pyramidal cells in the CA1 region are especially prone to oxidative damage [20,21]. Moreover, it is well established that various parts of the brain, including the hippocampus, are susceptible to ischaemia-related structural and functional damage [22]. Preclinical studies have shown that circulatory malfunctions lead to selective neuronal loss in CA1 pyramidal neurons, which impairs memory functions [23,24]. Given these factors, the CA1 region is a critical area for assessing the neuroprotective effects of our intervention, as it represents a key target for mitigating ischaemic damage and improving cognitive outcomes. Although the CA2, CA3, and DG regions are also of interest and have been studied in other contexts, our study specifically targeted CA1 due to its direct relevance to ischaemic damage and cognitive function.

In this study, we used an automated gait analysis system similar to the CatWalk system employed in previous research [25] to assess the effect of microcurrent therapy on post-stroke recovery in rats. This system enabled a precise and unbiased evaluation of locomotor activity, ensuring the reliability of our measurements, as validated in previous studies [26,27,28]. The results indicate that the control rats demonstrated normal mobility levels, developing a baseline for comparison. The rats that experienced stroke without subsequent treatment revealed significant reductions in both movement distance and speed, emphasising microcurrent therapy’s severe influence on motor function.

The groups treated with microcurrent therapy post-stroke showed marked improvements in movement. Immediate therapy post-stroke notably improved distance and speed, whereas pre- and post-treatment observations showed that their recovery levels approached those of the control group. These results indicate that microcurrent therapy effectively improves motor function recovery in stroke-afflicted rats. Furthermore, the use of a reliable motion analysis system, akin to the CatWalk system, proved pivotal in objectively quantifying recovery extent, underscoring microcurrent therapy’s potential as both a preventive and rehabilitative stroke treatment.

A histological analysis further verified the protective effects of microcurrent therapy. The total infarction area was significantly decreased in the post- and pre/post-treatment groups compared to the MCAO group. This reduction was consistent across various brain regions, including different bregma levels. An immunohistochemical analysis revealed elevated inflammatory markers (CD68, IL-6, and TNF-α) in the MCAO group, which were significantly reduced in the microcurrent-treated groups. The untreated MCAO group demonstrated high levels of these inflammatory markers, whereas the control group exhibited no inflammation. The post-treatment group showed significantly lower inflammatory marker levels compared to the MCAO group, and the pre/post-treatment group demonstrated even greater reductions, approaching the levels observed in the control group. This reduction in inflammation is crucial, as inflammation plays a significant role in secondary brain injury following ischaemia [10]. A previous study revealed that electrical stimulation modulates inflammatory responses, reducing proinflammatory cytokine and marker expression [29]. Our results indicate that microcurrent therapy effectively suppresses inflammation, contributing to its neuroprotective effects.

This study used a Western blot analysis to evaluate the effect of microcurrent therapy on inflammatory and apoptotic markers in an MCAO-induced stroke model. The untreated MCAO rats demonstrated elevated CD68, TNF-α, MMP8, NF-kB, IL-6, IL-1β, caspase-3, PARP, and BAX levels, indicating heightened inflammation and apoptosis compared to the controls. The rats receiving microcurrent therapy post-MCAO demonstrated significant reductions in these markers accompanied by decreased TLR4 and MyD88 expressions, indicating microcurrent therapy’s efficacy in mitigating cerebral damage [30].

The phosphorylation Chk1 and Chk2 levels were analysed via Western blotting to evaluate DNA damage. Group B demonstrated elevated p-Chk1 and p-Chk2 levels compared to group A. During microcurrent therapy, the p-Chk1 levels were similar between groups B and C, whereas p-Chk2 expression was reduced. Group D demonstrated levels comparable to group A, indicating the potential mitigation of ATM/ATR-mediated DNA damage responses that are crucial in cell cycle regulation and apoptosis [31]. These results underscore microcurrent therapy’s ability to modulate DNA damage pathways, contributing to neuroprotection in ischaemic stroke models.

Additionally, microcurrent therapy upregulated the angiogenic factor VEGF, promoting improved angiogenesis associated with improved histological and functional recovery. This result aligns with previous research emphasising electrical stimulation’s role in neural repair and recovery during early stroke stages [32]. Our results support its benefits in neurodegenerative conditions by improving vascular integrity and reducing arterial pathology, which is consistent with prior studies on microcurrent therapy [32].

Our study revealed that microcurrent therapy modulates specific signalling pathways crucial for inflammation and apoptosis. The TLR pathway involving TLR4 and MyD88 was notably downregulated in the microcurrent-treated groups. This pathway plays a pivotal role in mediating inflammatory responses and triggering proinflammatory cytokine production, such as TNF-α and IL-6, which induces apoptosis in chronic inflammatory conditions. Our results indicate that microcurrent therapy effectively suppresses TLR signalling, thereby reducing inflammation and apoptosis [13].

Furthermore, microcurrent therapy affected the MAPK signalling pathway, specifically influencing the ERK and p38 MAPK phosphorylation levels. The elevated phosphorylation levels of ERK and p38 MAPK observed in the MCAO group indicate increased inflammation and apoptosis. In contrast, microcurrent therapy significantly attenuated the phosphorylation of these proteins, indicating its role in mitigating these detrimental processes. This aligns with a previous study emphasising the neuroprotective potential of MAPK pathway modulation in ischaemic conditions [14].

The observed inflammatory marker reduction and neuronal tissue viability improvement in our study parallel the results obtained from Alzheimer’s disease models, where microcurrent therapy demonstrated significant neuroprotection by reducing inflammation and promoting neuronal repair [13,14]. These collective results underscore the therapeutic potential of microcurrent therapy in managing neuroinflammatory conditions through the targeted modulation of key signalling pathways.

In addition to the well-known mechanisms of ischaemic neuronal death—such as excitotoxicity, necrosis, apoptosis, pyroptosis, blood–brain barrier disruption, and neurovascular damage—microcurrent therapy may exert its neuroprotective effects through several other potential mechanisms. For example, microcurrent stimulation could enhance neurogenesis and synaptic plasticity by modulating the expression of neurotrophic factors like Brain-Derived Neurotrophic Factor (BDNF) [33]. Furthermore, it may influence mitochondrial function, reducing oxidative stress and promoting energy production in neurons. These effects could collectively contribute to the stabilisation of neural networks and support functional recovery after stroke. Expanding on these mechanistic insights could guide future research, helping to optimise microcurrent therapy protocols and potentially uncover new therapeutic targets. 

This study has several limitations that should be addressed despite the promising results obtained. The relatively small sample size limits the statistical power and generalizability of the results. Larger studies are required to confirm these results and ensure their applicability to broader populations. Additionally, this study focused solely on short-term outcomes after microcurrent therapy. The long-term effects and potential side effects of prolonged microcurrent therapy were not evaluated, which is crucial for evaluating the overall treatment safety and efficacy. Another limitation is the lack of mechanistic insight into how microcurrent therapy exerts its effects. Our results indicate that microcurrent therapy modulates inflammation, angiogenesis, oxidative stress, and apoptosis, but the precise molecular pathways involved remain unclear. Further research is warranted to elucidate these mechanisms and determine potential targets for improving the therapeutic efficacy of microcurrent therapy. Moreover, this study utilised a single animal model (Sprague Dawley rats) of MCAO. The applicability of these results to other models or species, including humans, warrants further investigation.

## 4. Materials and Methods

### 4.1. Experimental Design: Group Allocation and the Induction of the MCAO Stroke Model

The Institutional Animal Care and Use Committee of the Catholic University of Daegu School of Medicine, in compliance with their established animal welfare regulations, approved the experimental procedures (IRB No. DCIAFCR-230524-12-Y). Twenty 8-week-old male Sprague Dawley rats weighing 300–330 grammes were acquired from Hyo-Chang Science (Daegu, Republic of Korea). The rats were kept under controlled environmental conditions with a steady temperature of 23 °C ± 2 °C, humidity levels of 45% ± 10%, and a 12 h light/dark cycle. Access to a standard rodent diet and water was provided ad libitum. The rats were categorised into four groups after a week-long adjustment period, with each group consisting of five animals.

Group A (control) served as the healthy control without any intervention. Group B (disease) included rats subjected to the middle cerebral artery occlusion (MCAO) model to simulate stroke conditions. Group C (treatment post-MCAO) received microcurrent therapy immediately after MCAO induction, which was continued for one week to evaluate its effectiveness in treating stroke symptoms. Group D (prevention and recovery) received microcurrent therapy both one week before and one week after MCAO induction. This dual-phase treatment aimed to evaluate the therapy’s capacity for both preventing cerebral damage and aiding in recovery post-stroke (Figure 9).

The MCAO/reperfusion (MCAO/R) model was developed following the intraluminal vascular occlusion technique outlined as previously described [34]. Initially, a mixture of 5% of isoflurane, 70% of nitrous oxide, and 30% of oxygen was used for anaesthesia, maintained with a continuous flow of 3% isoflurane. The left common carotid artery, internal carotid artery (ICA), and external carotid artery (ECA) were surgically exposed. The ECA was then cut and a 4-0 monofilament nylon suture with a coated rounded tip (filament size: 4-0, diameter: 0.19 mm, tip diameter: 0.35 ± 0.02 mm, suture length: 30 mm, tip length: 5–6 mm; sourced from LMS Korea, Seongnam, Republic of Korea) was inserted through the ECA stump up to the ICA and MCA bifurcation. The animals were re-anaesthetised for reperfusion after two hours of occlusion, initiated by withdrawing the suture to restore ICA blood flow. The ECA was permanently sealed off. Throughout the procedure, the animal’s body temperature was controlled using a heating pad.

### 4.2. Microcurrent Therapy Protocol 

Microcurrent therapy was administered for 12 h daily during specific periods defined for each experimental group to evaluate its effects (Figure 9). This regimen was consistently maintained for one or two weeks according to the group. The microcurrent was delivered through a wire that was connected from the generator (Ecure, Busan, Republic of Korea) to a copper plate, the dimensions of which matched the cage floor. This arrangement ensured that the mice absorbed the current through their feet, effectively channelling the therapy to their brains (Figure 9).

The therapy for the nocturnal mice was applied during their night cycle to align with their natural active phase, considering that microcurrent therapy in humans is typically applied during the daytime to coincide with their active hours. The research by Kim et al. [33] revealed that different microcurrent waveforms are beneficial, with the step form waveform being particularly effective in improving clinical outcomes, including cognitive abilities and protein synthesis associated with Alzheimer’s disease in mice. Accordingly, a step form waveform at settings of 0, 1.5, 3, and 5 V with waveform superposition was selected. The therapy parameters were set at an intensity of 1 μA (250 ohm), a voltage of 5 V, and a base frequency of 7 Hz, which was improved by an additional frequency of 44 kHz.

### 4.3. Behavioural Motion Analysis 

The rats were first acclimatised to the open-field environment for 30 min to begin the behavioural evaluation. This habituation phase was crucial to minimise stress and encourage natural exploratory behaviour. Afterwards, the actual analysis commenced with each rat being allowed to roam freely in a delineated 90 cm by 90 cm arena for 5 min. Movement within this area was precisely monitored using a high-resolution camera system equipped with the SMART 3.0 video-tracking software (Panlab, Barcelona, Spain). This setup was designed to capture and analyse the key locomotion parameters, such as the total walking distance and average speed, ensuring accurate and reliable behavioural data.

### 4.4. Gross Morphology of Brain Tissue Post-MCAO among Groups

We captured gross photographs of brain tissue affected by left MCAO from three distinct perspectives to compare brain tissue necrosis across different conditions. The first image provides a lateral view, emphasising the rounded contours typical of the cerebral hemispheres. This perspective is useful both for illustrating the undisturbed structure in a controlled setting and for indicating the initial effect of MCAO. The second image illustrates a superior view, highlighting the natural segmentation and detailed morphology of the brain’s lobes. This view is crucial for pinpointing regions that may be impacted by ischaemic damage. The third image, an oblique view, explores the dimensional depth and structural integrity of the brain, potentially determining deformation or other alteration areas caused by occlusion.

### 4.5. Tissue Preparation and Brain Sectioning for Histological and Immunohistochemical Analysis

The rats were euthanised to harvest brain tissues for analysis after completing a week of microcurrent treatment followed by behavioural testing. Each rat was perfused intracardially with saline to clear blood from the cerebral vessels, immediately followed by fixation with 10% neutral buffered formalin (NBF) for histological examination. The whole brain was carefully extracted and post-fixed by immersion in 10% NBF after perfusion to ensure thorough tissue morphology preservation.

The brains designated for Western blotting were prepared separately. They were frozen and stored at −80 °C immediately after removal to preserve protein integrity.

The brains were subsequently sectioned for histological staining, including haematoxylin and eosin (H&E), cresyl violet (CV), and immunohistochemistry (IHC). The brains were sliced in the coronal plane at specific bregma points using a brain matrix: +1 ± 0.5 mm, −2 ± 0.5 mm, and −5 ± 0.5 mm. This procedure divided the brain into four distinct segments, enabling a detailed examination of the regional anatomical features and pathological changes. The brain segments were embedded in paraffin and prepared in coronal sections of 5 μm in thickness. Histopathological changes in the hippocampus CA1 region were evaluated with H&E and CV staining. For cell counting in the CA1 region, 5 areas per slide were analysed to obtain an average cell count per slide. The number of cells in a 1 mm^2^ area of the CA1 region was counted, and changes in the cell structure were analysed. The data from each animal were collected from multiple slides, and the exact number of slides per animal used for cell counting is reported in the Results Section. In Figure 3C, approximately 44–45 plots are shown for group B. Each plot represents a specific area of the CA1 region. The data from 5 rats used for histological evaluation were obtained from multiple slides, with an average of 9 data points acquired per animal. The mean value was based on cell counting from 5 distinct areas per slide. Axiophot Photomicroscope (Carl Zeiss, Oberkochen, Germany) and AxioCam MRc5 (Carl Zeiss) were used to examine the prepared sections and store images, and Axio-Vision SE64 (Carl Zeiss) was utilised to analyse the stored images.

The cell count data are reported as the number of cells per mm^2^ based on a 1 mm × 1 mm area of each slide. Any discrepancies in units were corrected to ensure an accurate representation of the results.

#### 4.5.1. Evaluation of Brain Infarction Area

To assess the cerebral infarction area, the brain sections were coronally sliced at specific bregma co-ordinates (+1 ± 0.5 mm, −2 ± 0.5 mm, and −5 ± 0.5 mm). Measurements of the infarcted area were taken at each of these points. The infarction areas from these regions were summed to obtain the total infarction area.

The infarction area as a percentage of the total area of the left hemisphere was calculated using the following formula:Cerebral Infarction Area (%) = (Total Infarction Area)/(Total Area of the left Hemisphere) × 100%

Note: The Total Infarction Area represents the sum of the infarction areas from the three measured regions. The Total Area of the left Hemisphere includes both infarcted and non-infarcted tissue.

The infarction area of the H&E-stained brain sections was measured and analysed five times with AxioVision SE64. This method provides a detailed measure of the extent of brain damage, enabling a precise evaluation of the ischaemic injury caused by the experimental conditions.

#### 4.5.2. Immunohistochemistry Protocol

The slides were initially rinsed in phosphate-buffered saline (PBS) for an IHC analysis. The sections were incubated in citrate buffer (pH: 6.0) for 30 min at 95 °C for antigen retrieval. After incubation, endogenous peroxidases were blocked with 0.3% of hydrogen peroxide in PBS for 30 min. Blocking for nonspecific antibody binding was performed in PBS with 10% of normal goat serum or normal horse serum (Vector Laboratories, Newark, CA, USA) for 30 min. The sections were then washed thrice with PBS and incubated with primary antibodies (1:100–1:200) at room temperature for 2 h. The following primary antibodies were used: rabbit anti-CD68 polyclonal antibody (ab125212; Abcam, Cambridge, UK), mouse anti-interleukin (IL)-6 monoclonal antibody (ab9324; Abcam), and mouse anti-tumour necrosis factor (TNF)-α monoclonal antibody (ab220210; Abcam). Subsequently, the sections were incubated with secondary antibodies (1:100), biotinylated anti-rabbit IgG or biotinylated anti-mouse IgG (Vector Laboratories), at room temperature for 1 h. The sections were washed with PBS thrice and treated with avidin–biotin–peroxidase complex (Vector Laboratories) for 1 h. The sections were washed again in PBS thrice and underwent a peroxidase reaction with 0.05 M Tris-HCl (pH: 7.6) containing 0.01% of hydrogen peroxide and 0.05% of 3,3′-diaminobenzidine (DAB; Sigma-Aldrich, St. Louis, MO, USA). Counterstaining with haematoxylin was performed. The slides were then evaluated and imaged using an Axiophot Photomicroscope (Carl Zeiss, Oberkochen, Germany) and AxioCam MRc5 (Carl Zeiss, Oberkochen, Germany).

#### 4.5.3. Immunohistochemistry Analysis

An experienced anatomist with 20 years of expertise who was blinded to the experimental groups conducted a histologic evaluation. The immunostained slides were imaged and analysed using an Axiophot Photomicroscope and AxioCam MRc5. The area of immunopositive staining in each group was measured and analysed five times using AxioVision SE64. The number and areas of immunopositive stained cells in each group were measured using AxioVision SE64 (Carl Zeiss, Oberkochen, Germany) in five random areas, with each area having a size of 1 mm × 1 mm.

### 4.6. Western Blot Analysis

A tissue sample measuring 0.2 × 0.2 × 0.2 cm was homogenised in 180 μL of 1× radioimmunoprecipitation assay buffer, which included 1× phosphate-buffered saline, 1% of NP–40, 0.5% of sodium deoxycholate, 0.1% of SDS, 10 μg/mL of phenylmethanesulfonyl fluoride, and a protease inhibitor cocktail tablet. The homogenised proteins were then extracted and quantified at 40 μg using a bicinchoninic acid assay kit (Thermo Fisher Scientific Inc., Waltham, MA, USA).

The proteins were separated through SDS–polyacrylamide gel electrophoresis on NuPAGE 4–12% bis-Tris gels (Invitrogen, Waltham, MA, USA) and transferred to a 0.2 μm section of polyvinylidene difluoride membrane (GE Healthcare Life Sciences, Amersham, Bucks, Germany). The membrane was blocked with a casein blocking buffer (Sigma-Aldrich Corp., St. Louis, MO, USA) and washed with phosphate-buffered saline with Tween^®^-20 (PBST, Bio-Rad Laboratories Inc., Hercules, CA, USA).

The solution was removed after blocking for 2 h in 5% casein blocking buffer, and the membrane was incubated with primary antibodies, including CD68 (1:500, ab125212; Abcam), IL-6 (1:500, ab9324; Abcam), IL-1β (1:500, ab18329; Abcam), TNF-α (1:500, sc-52746; Santa Cruz Biotechnology, Santa Cruz, CA, USA), MyD88 (1:500, sc-74532; Santa Cruz Biotechnology), TLR4 (1:500, sc-293072; Santa Cruz Biotechnology), vascular endothelial growth factor (VEGF) (1:500, sc-7269; Santa Cruz Biotechnology), matrix metalloproteinase 8 (MMP8) (1:500, LS-C490399; LifeSpan BioSciences, Seattle, WA, USA), nuclear factor kappa-light-chain-enhancer of activated B cells (NF-kB) (1:500, #8242; Cell Signaling Technology, Danvers, MA, USA), cleaved caspase-3 (1:500, #9664; Cell Signalling), caspase-3 (1:500, #14,220; Cell Signalling), cleaved poly (ADP-ribose) polymerase (PARP) (1:500, #9541; Cell Signalling), PARP (1:500, #9542; Cell Signalling), BAX (1:500, #5023; Cell Signalling), Bcl-2 (1:500, #2870; Cell Signalling), protein extracellular signal-regulated kinase (p-ERK) (1:500, #9101; Cell Signalling), ERK (1:500, #9102; Cell Signalling), p-p38 MAPK (1:500, #9211; Cell Signalling), p38 MAPK (1:500, #8690; Cell Signalling), p-Chk1 (1:250, #2348; Cell Signalling), Chk1 (1:500, sc-8408; Santa Cruz Biotechnology), p-Chk2 (1:250, #2197; Cell Signalling), Chk2 (1:500, sc-17748; Santa Cruz Biotechnology), and β-actin (1:1000, A2228; Sigma-Aldrich Corp.), all diluted in 5% casein blocking buffer. The membrane was incubated with these primary antibodies at 4 °C for 24 h.

The primary antibody solution was removed after incubation, and the membrane was washed thrice with PBST. The secondary antibody, diluted in 5% casein blocking buffer, was applied to the membrane and incubated at room temperature for 1 h. The membrane was then washed thrice for 15 min with PBST. The protein bands were visualised using ECL Western Blotting Substrate and X-ray film development. Protein quantification was conducted using GraphPad Prism (Version 9.5.1; GraphPad Software Inc., San Diego, CA, USA). β-Actin was used as a loading control to normalise the protein levels across different samples. Using this approach ensured that any variations in protein loading and transfer efficiency were accounted for, allowing for an accurate comparison of the target protein expression levels.

### 4.7. Statistical Analysis

We conducted a pilot study to calculate the sample size, with the primary endpoint being the infarction area. In the pilot study, we used one rat in each group and evaluated five randomly selected fields in each group. The effect size was 0.49 and, using an analysis of variance (ANOVA) with a significance level of 0.05, we determined that at least 76 fields were needed to achieve a power of at least 95% for this effect size. Since five fields can be obtained from one rat, 16 rats were required. Considering a 20% drop rate, we determined the final sample size as 20. Additionally, we performed a post hoc power analysis using G*Power 3 (Heinrich-Heine-University, Dusseldorf, Germany), which showed a power of 1.00. A statistical evaluation was conducted using the Statistical Package for the Social Sciences software, version 25.0, for Windows (SPSS Inc., Chicago, IL, USA). Alongside basic descriptive statistics (averages and standard deviations), the ANOVA technique was used to investigate differences both within and between groups. Tukey’s test was used for further post hoc comparisons if the ANOVA results revealed significant variances among groups. Average values were reported alongside 95% confidence intervals and all the results are presented as means ± standard errors. Statistical significance was determined using *p*-values, with thresholds set at * *p* < 0.05, ** *p* < 0.01, *** *p* < 0.001, and **** *p* < 0.001 to indicate varying significant levels. Following this study, a post hoc power analysis was conducted, indicating an achieved power of >0.95.

## 5. Conclusions

Our study confirms the growing body of evidence that microcurrent therapy is a promising treatment modality for ischaemic brain injury. Microcurrent therapy has potential as a noninvasive therapeutic strategy for stroke rehabilitation by reducing ischaemic damage, modulating inflammatory responses, and promoting functional recovery. However, further research is required to address the study limitations, optimise treatment parameters, and fully elucidate the underlying mechanisms of action. This will be crucial for advancing microcurrent therapy towards clinical application and improving outcomes for patients with ischaemic stroke.

## Figures and Tables

**Figure 1 ijms-25-10034-f001:**
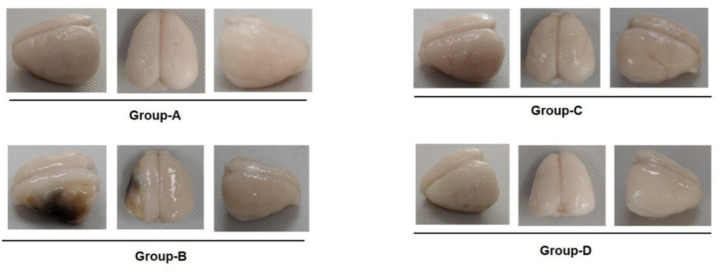
A morphological comparison of brain tissue among experimental groups A–D. This figure displays brain tissues from four groups: group A (control), with an intact morphology; B (disease), showing significant disruption and necrosis due to MCAO; and C (treatment post-MCAO) and D (prevention and recovery), both of which indicate signs of morphological recovery, emphasising the effectiveness of microcurrent therapy in mitigating ischaemic damage and promoting structural restoration.

**Figure 2 ijms-25-10034-f002:**
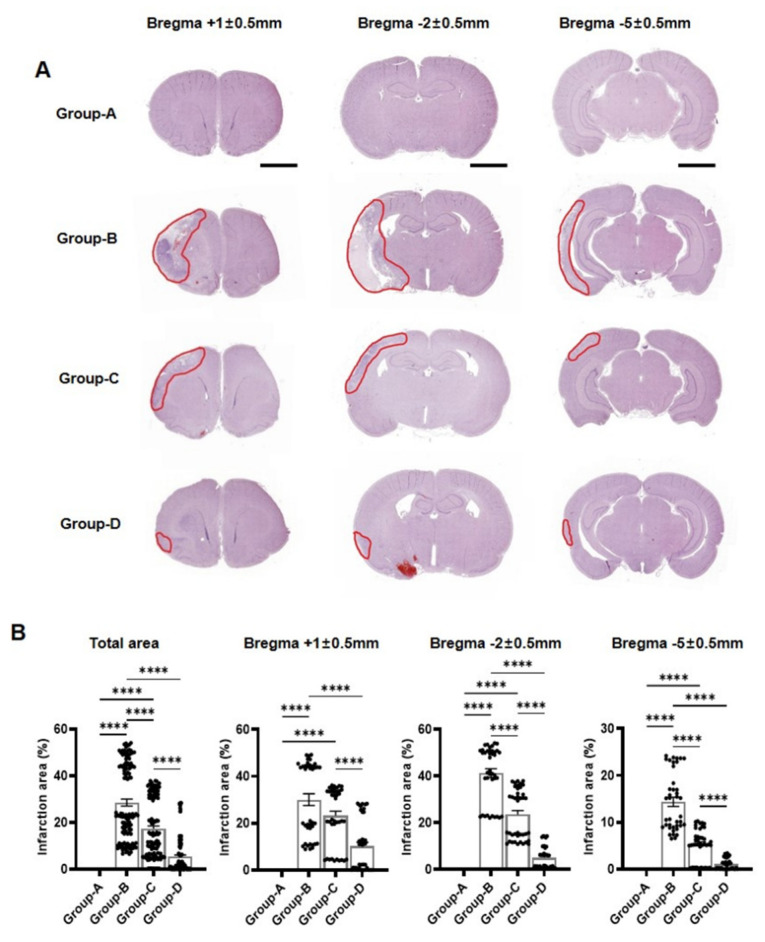
Histological analysis and infarction area quantification in groups A–D. (**A**) The images display stained brain slices at bregma +1 ± 0.5 mm, −2 ± 0.5 mm, and −5 ± 0.5 mm levels. Infarction areas were delineated with a red circle in the figure. (**B**) The infarction areas for these groups across total, bregma +1 ± 0.5 mm, −2 ± 0.5 mm, and −5 ± 0.5 mm levels. Scale bar = 3 mm. Group A (control); group B (disease); group C (treatment post-MCAO); group D (prevention and recovery). **** *p* indicates < 0.001 (post hoc testing between groups).

**Figure 3 ijms-25-10034-f003:**
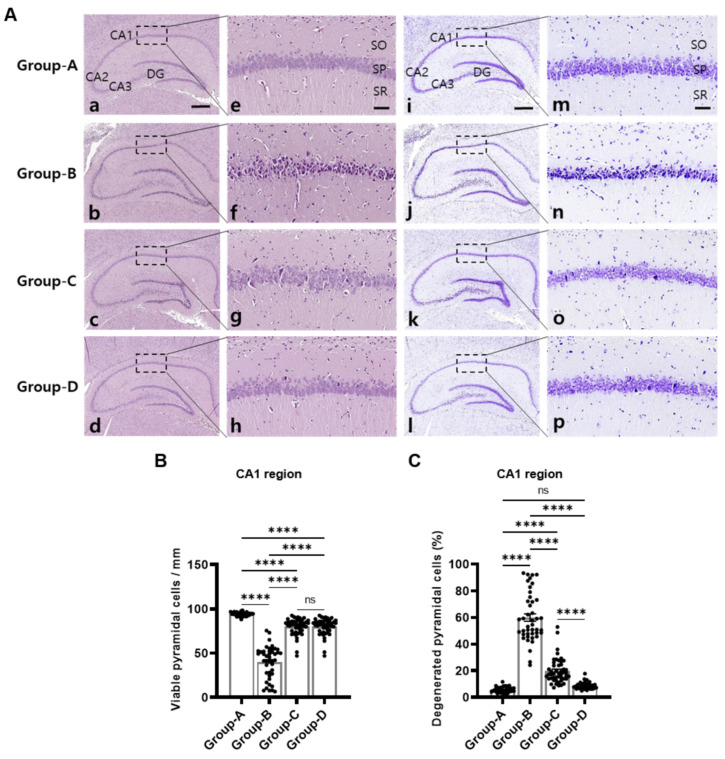
A histological examination and quantitative analysis of the hippocampal regions in groups A–D. H&E staining (**a**–**h**) and cresyl violet (CV) staining of rat hippocampus (**i**–**p**). (**a**–**d**) and (**i**–**l**) show hippocampal slices from groups A through D, each stained to highlight regions, including CA1, CA2, CA3, and DG. (**e**–**h**) and (**m**–**p**) provide magnified views of the CA1 region for each group, focusing on pyramidal cell integrity (**A**). Viable pyramidal cells per mm in the CA1 area and a statistical analysis reveal significant variations among the groups (**B**). The percentage of degenerative pyramidal cells in a 1 mm range of the CA1 region from groups A through D (**C**). CA: cornu ammonis; DG: dentate gyrus; SO: stratum oriens; SP: stratum pyramidale; SR: stratum radiatum. Scale bar = 400 μm (**a**–**d**,**i**–**l**); 50 μm (**e**–**h**,**m**–**p**). Group A (control); group B (disease); group C (treatment post-MCAO); group D (prevention and recovery). **** *p* indicates < 0.001 (post hoc testing between groups).

**Figure 4 ijms-25-10034-f004:**
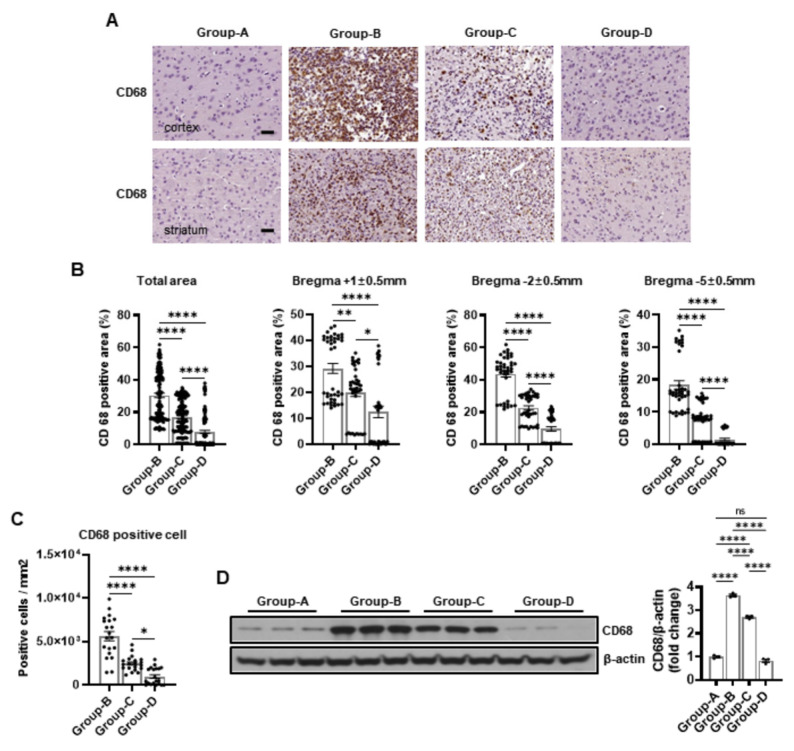
Immunohistochemical and Western blot analyses of CD68 expression in groups A–D. (**A**) Representative images of CD68 immunostaining in the cortex and striatum for each group. (**B**) The percentage of CD68-positive area in different brain regions. (**C**) The number of CD68-positive cells per mm^2^ in the cortex and striatum for each group. (**D**) The Western blot results for CD68. Scale bar = 50 μm. Group A (control); group B (disease); group C (treatment post-MCAO); and group D (prevention and recovery). * *p* indicates < 0.05, ** *p* indicates < 0.01, and **** *p* indicates < 0.001 (post hoc testing between groups).

**Figure 5 ijms-25-10034-f005:**
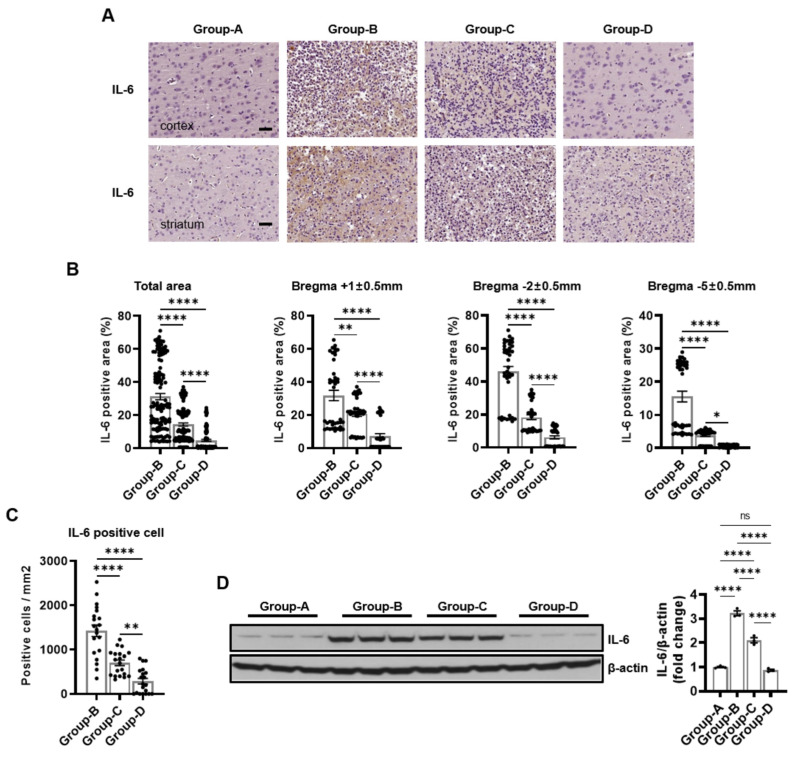
Immunohistochemical and Western blot analyses of IL-6 expression in groups A–D. (**A**) Representative immunostaining images of IL-6 in the cortex and striatum in each group. (**B**) The percentage of IL-6-positive areas in the cortex and striatum. (**C**) The number of IL-6-positive cells per mm^2^ in the cortex and striatum in each group. (**D**) The Western blot results of IL-6. Scale bar = 50 μm. Group A (control); group B (disease); group C (treatment post-MCAO); and group D (prevention and recovery). * *p* indicates < 0.05, ** *p* indicates < 0.01, and **** *p* indicates < 0.001 (post hoc testing between groups).

**Figure 6 ijms-25-10034-f006:**
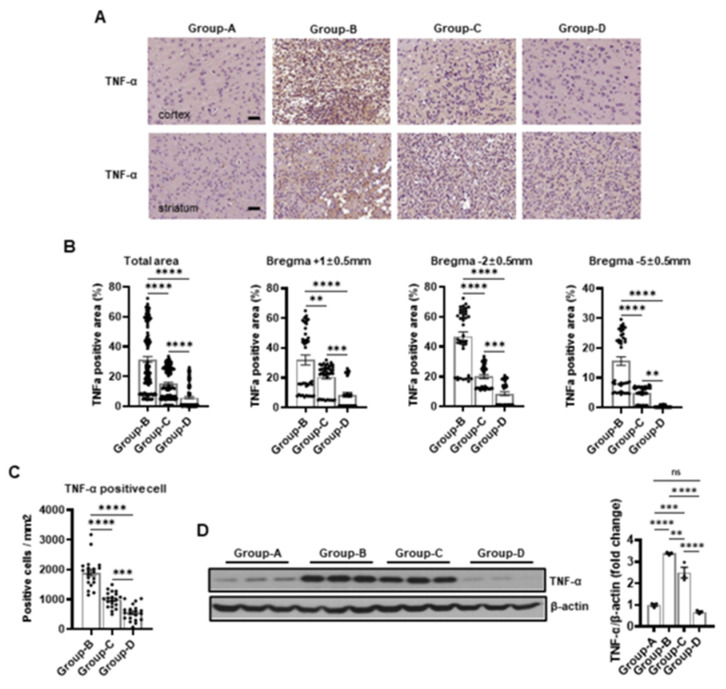
Immunohistochemical and Western blot analyses of TNF-α expression in groups A–D. (**A**) Representative immunostaining images showing TNF-α expression in the cortex and striatum for each group. (**B**) The TNF-α-positive area in the cortex and striatum. (**C**) The count of TNF-α-positive cells per mm^2^ in the cortex and striatum. (**D**) The Western blot results for TNF-α. Scale bar = 50 μm. Group A (control); group B (disease); group C (treatment post-MCAO); and group D (prevention and recovery). ** *p* indicates < 0.01, *** *p* indicates < 0.001, and **** *p* indicates < 0.001 (post hoc testing between groups).

**Figure 7 ijms-25-10034-f007:**
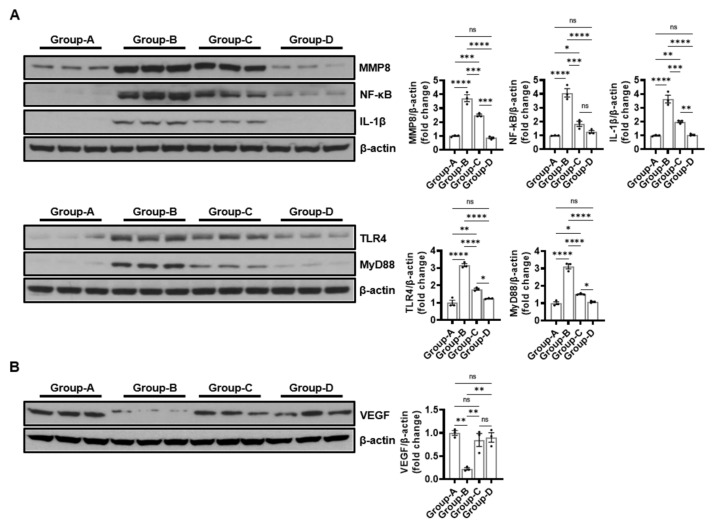
The Western blot results of immune-related proteins (**A**) and angiogenic factors (**B**). (**A**) MMP8, NF-kB, IL-1β, TLR4, and MyD88 in groups A–D (**B**) VEGF. Group A (control); group B (disease); group C (treatment post-MCAO); and group D (prevention and recovery). * *p* indicates < 0.05, ** *p* indicates < 0.01, *** *p* indicates < 0.001, and **** *p* indicates < 0.001 (post hoc testing between groups).

**Figure 8 ijms-25-10034-f008:**
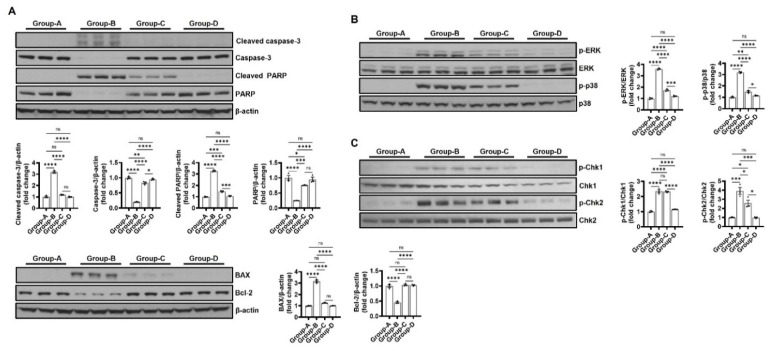
The Western blot results of (**A**) apoptosis-related proteins (caspase-3, cleaved-PARP, BAX, and Bcl-2), (**B**) MAPK proteins (ERK and p38), and (**C**) DNA-damage-related proteins (Chk1 and Chk2 phosphorylation). Group A (control); group B (disease); group C (treatment post-MCAO); and group D (prevention and recovery). * *p* indicates < 0.05, ** *p* indicates < 0.01, *** *p* indicates < 0.001, and **** *p* indicates < 0.001 (post hoc testing between groups).

**Figure 9 ijms-25-10034-f009:**
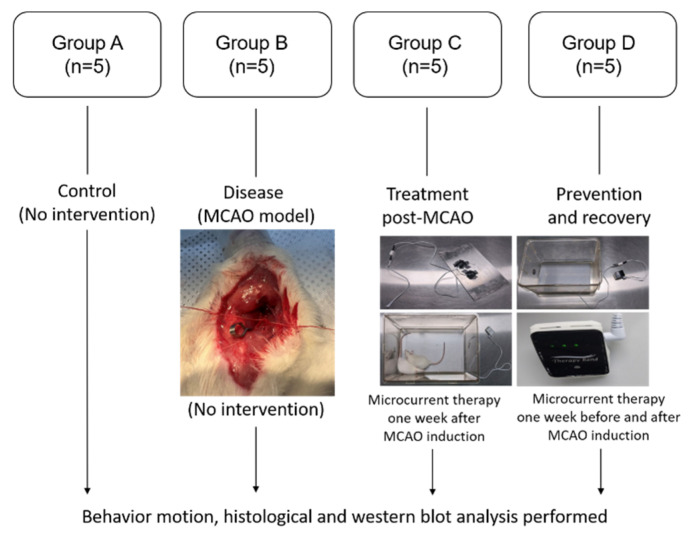
An illustration of the study protocol. Thirty-two rats were randomly allocated into four groups: group A (control), serving as the healthy control without any intervention; group B (disease), including rats subjected to the MCAO model to simulate stroke conditions; group C (treatment post-MCAO), including rats receiving microcurrent therapy immediately after MCAO induction, which was continued for one week to evaluate its effectiveness in treating stroke symptoms; and group D (prevention and recovery), including rats receiving microcurrent therapy both one week before and one week after MCAO induction. MCAO: middle cerebral artery occlusion.

## Data Availability

All data generated or analysed during this study are included in this published article.

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
