# Peer review of "The Neuroprotective Effects of Peripheral Nerve Microcurrent Stimulation Therapy in a Rat Model of Middle Cerebral Artery Occlusion"

_ijms, 2024, doi:10.3390/ijms251810034_

Round 1
Reviewer 1 Report
Comments and Suggestions for Authors
1. General remarks:
The authors achieved an interesting, laborious and topic – as long as actually, it is unfortunately, well-known that for the CNS lesions there is practically no cure – experimental study
2. Specific remarks and suggestions:
- the authors should explain, including with better/ more edifying references how the microcurrents – applied peripherally, on the femoral nerve, so off/ far from the brain – might action long-distance, intimately, i. e. on cellular/ molecular targets, in the experimental MCAO infarction stroke zones [”It modulates neuroinflammation via mitogen-activated 64 protein kinase (MAPK) and Toll-Like Receptor 4 (TLR4) signalling pathways, potentially reducing neuroinflammatory proteins and improving ischaemic stroke outcomes” and ”the mice absorbed the current through their feet, effectively channelling – how ? (o. n.) – the therapy to their brains)] ?
- the ”Materials and Methods” section is – unusually – placed after the ”Discussion” section, including with this study’s conclusion (? !)
Author Response
Dear Editors:
We thank you for the opportunity to revise our manuscript. Additionally, we appreciate the time you have dedicated to reviewing our manuscript.
The comments were greatly helpful in improving the contents and revising the errors in our manuscript.
We did our best to revise the paper according to the referees’ comments. For convenience, we itemized the referees’ original comments and put the corresponding corrections/rebuttal immediately after each comment. The modified parts have been marked in red within the text.
We believe that the quality of the manuscript is now much improved and hope that the revised paper is suitable for publication in International Journal of Molecular Sciences.
Thank you for your consideration.
Sincerely yours,
Dong Rak Kwon, MD, PhD
Department of Rehabilitation Medicine
Catholic University of Daegu School of Medicine
33 Duryugongwon-ro 17-gil, Nam-Gu, Daegu, Korea, 705-718
Phone: +82 53 650 4687 Fax: +82 53 622 4687
E-mail: coolkwon@cu.ac.kr
Point-by-Point Responses to Editor
Manuscript ID: ijms-3150070
Title: Neuroprotective effects of peripheral nerve microcurrent stimulation therapy in a rat model of middle cerebral artery occlusion
Authors: Yoon-Jin Lee1†, Eun Sang Kwon2†, Yong Suk Moon3, Jeong-Rang Jo4 and Dong Rak Kwon4*
Affiliation: 1. Department of Biochemistry, College of Medicine, Soonchunhyang University, Cheonan 31151, Republic of Korea
- Department of Medicine, College of Medicine, Keimyung University, Daegu 42601, Republic of Korea
- Department of Anatomy, Catholic University of Daegu School of Medicine, Daegu 42472, Republic of Korea
- Department of Rehabilitation Medicine, Catholic University of Daegu School of Medicine, Daegu 42472, Republic of Korea
* Correspondence:
Dong Rak Kwon, MD, PhD
Department of Rehabilitation Medicine, Catholic University of Daegu School of Medicine, 33 Duryugongwon-ro 17-gil, Nam-Gu, Daegu, 42472, Korea
Telephone: 82-53-650-4878
Fax: 82-53-622-4687
|
Reviewer 1 1. General remarks:
The authors achieved an interesting, laborious and topic – as long as actually, it is unfortunately, well-known that for the CNS lesions there is practically no cure – experimental study
2. Specific remarks and suggestions:
Comment 1: the authors should explain, including with better/ more edifying references how the microcurrents – applied peripherally, on the femoral nerve, so off/ far from the brain – might action long-distance, intimately, i. e. on cellular/ molecular targets, in the experimental MCAO infarction stroke zones [”It modulates neuroinflammation via mitogen-activated 64 protein kinase (MAPK) and Toll-Like Receptor 4 (TLR4) signalling pathways, potentially reducing neuroinflammatory proteins and improving ischaemic stroke outcomes” and ”the mice absorbed the current through their feet, effectively channelling – how ? (o. n.) – the therapy to their brains)] ?
Response 1: Thank you for your valuable feedback. We added as follows: The mechanisms through which microcurrent therapy exerts its effects are not fully elucidated, but several hypotheses are supported by existing research. Microcurrent therapy is believed to enhance cellular energy production, improve amino acid transport, and stimulate protein synthesis, which can collectively reduce inflammation and support tissue repair [9]. Persistent neuroinflammation is a significant factor in neurodegenerative diseases and cerebrovascular events, such as stroke [10]. When applied peripherally, such as to the femoral nerve, microcurrent therapy may initiate systemic responses through both humoral and neural pathways. For instance, stimulation of the femoral nerve could lead to the release of blood-borne factors, including small molecules and neurotrophic factors that travel to the central nervous system [11]. Additionally, neural pathways might be activated, potentially modulating central nervous system processes indirectly [12]. While the exact pathways remain under investigation, current evidence suggests that these mechanisms could contribute to neuroprotection and modulation of neuroinflammation in stroke models.
Comment 2: the ”Materials and Methods” section is – unusually – placed after The ”Discussion” section, including with this study’s conclusion (? !)
Response 2: Thank you for your valuable feedback and insightful comments on our manuscript. We have formatted the manuscript according to the guidelines of the International Journal of Molecular Sciences. The "Materials and Methods" section is placed after the "Discussion" as required by the journal, and the conclusion is placed at the very end.
|
Reviewer 2 Report
Comments and Suggestions for Authors
The manuscript “Neuroprotective effects of peripheral nerve microcurrent stimulation therapy in a rat model of middle cerebral artery occlusion” by Yoon-Jin Lee et al. is devoted to study the neuroprotective effects of peripheral nerve microcurrent stimulation therapy in a rat middle cerebral artery occlusion model.
The manuscript is written in good scientific language, but it requires major revision and answers to questions before it is published in the International Journal of Molecular Sciences.
1) Figure 2. The authors need to present in the figures a control group - group A. This is necessary for correct data comparison.
2) When presenting data analyzed using statistical methods (in particular, in the figures 3, 4, 5, 6, 7, 8), the authors do not indicate anywhere by which parameters the data were calculated, which P the characters "*" correspond to. It is necessary to fix this.
3) Why was the CA1 site selected for analysis? Have the CA2, CA3 and DG sites been studied?
4) How was the damage area calculated? It is necessary to present the manuscript in the main section of the results.
5) Line 123. Are the authors really referring to Figure 2 and not Figure 5?
6) Line 161. What was the stain done?
7) Figures 4, 5 and 6. Each letter requires mandatory decoding. It is necessary to explain what is represented under the letters.
8) Figure 7. The authors need to explain and provide relevant information in the text of the manuscript, why was β-actin used?
Figure 8. Similar to the previous remark: β-actin, p38 and Chk2.
I would also like to recommend that in the future the authors take a more responsible attitude to the design of the manuscript text - currently, the manuscript has an extremely low quality of design, which seriously complicates the analysis of the work:
1) Lines 76-69. Incorrect use of text alignment in the center.
2) In some places, the authors neglect the accuracy of using the "%" characters, duplicating them, in particular: lines 174-175. It is necessary to check the entire text.
3) In some cases, the authors neglect or use excessive amounts of space characters. In particular, lines 164, 217.
4) Careless design of Figure 8 - a huge piece of blank page between A and B is incomprehensible.
5) Extremely poor quality of drawings - it is necessary to submit images according to the requirements of the IJMS.
I sincerely hope that the comments and recommendations I have made will be useful to the team of highly respected authors and will be able to help improve the quality of the manuscript they submit. I wish you great success!
Author Response
Dear Editors:
We thank you for the opportunity to revise our manuscript. Additionally, we appreciate the time you have dedicated to reviewing our manuscript.
The comments were greatly helpful in improving the contents and revising the errors in our manuscript.
We did our best to revise the paper according to the referees’ comments. For convenience, we itemized the referees’ original comments and put the corresponding corrections/rebuttal immediately after each comment. The modified parts have been marked in red within the text.
We believe that the quality of the manuscript is now much improved and hope that the revised paper is suitable for publication in International Journal of Molecular Sciences.
Thank you for your consideration.
Sincerely yours,
Dong Rak Kwon, MD, PhD
Department of Rehabilitation Medicine
Catholic University of Daegu School of Medicine
33 Duryugongwon-ro 17-gil, Nam-Gu, Daegu, Korea, 705-718
Phone: +82 53 650 4687 Fax: +82 53 622 4687
E-mail: coolkwon@cu.ac.kr
Point-by-Point Responses to Editor
Manuscript ID: ijms-3150070
Title: Neuroprotective effects of peripheral nerve microcurrent stimulation therapy in a rat model of middle cerebral artery occlusion
Authors: Yoon-Jin Lee1†, Eun Sang Kwon2†, Yong Suk Moon3, Jeong-Rang Jo4 and Dong Rak Kwon4*
Affiliation: 1. Department of Biochemistry, College of Medicine, Soonchunhyang University, Cheonan 31151, Republic of Korea
- Department of Medicine, College of Medicine, Keimyung University, Daegu 42601, Republic of Korea
- Department of Anatomy, Catholic University of Daegu School of Medicine, Daegu 42472, Republic of Korea
- Department of Rehabilitation Medicine, Catholic University of Daegu School of Medicine, Daegu 42472, Republic of Korea
* Correspondence:
Dong Rak Kwon, MD, PhD
Department of Rehabilitation Medicine, Catholic University of Daegu School of Medicine, 33 Duryugongwon-ro 17-gil, Nam-Gu, Daegu, 42472, Korea
Telephone: 82-53-650-4878
Fax: 82-53-622-4687
|
Reviewer 2 The manuscript “Neuroprotective effects of peripheral nerve microcurrent stimulation therapy in a rat model of middle cerebral artery occlusion” by Yoon-Jin Lee et al. is devoted to study the neuroprotective effects of peripheral nerve microcurrent stimulation therapy in a rat middle cerebral artery occlusion model.
The manuscript is written in good scientific language, but it requires major revision and answers to questions before it is published in the International Journal of Molecular Sciences.
Comment 1) Figure 2. The authors need to present in the figures a control group - group A. This is necessary for correct data comparison. Response 1) Thank you for your valuable feedback on our manuscript. Regarding your comment on Figure 2, we acknowledge the importance of including a control group for accurate data comparison. We will revise Figure 2 to incorporate Group A as the control group, as suggested. This addition will help ensure a more robust comparison and strengthen the validity of our findings. Thank you for pointing this out. We will make the necessary adjustments and update the figure accordingly.
Comment 2) When presenting data analyzed using statistical methods (in particular, in the figures 3, 4, 5, 6, 7, 8), the authors do not indicate anywhere by which parameters the data were calculated, which P the characters "*" correspond to. It is necessary to fix this. Response 2) Thank you for your valuable feedback on our manuscript. In response to your comment regarding the presentation of statistical data in Figures 3, 4, 5, 6, 7, and 8, we will make the following revisions:
of the significance levels in the figure legends. The revised notation will be as follows: *p indicates < 0.05, **p indicates < 0.01, ***p indicates < 0.001 and ****p indicates < 0.001 (post-hoc testing between groups)
Comment 3) Why was the CA1 site selected for analysis? Have the CA2, CA3 and DG sites been studied? Response 3) Thank you for your insightful question regarding the selection of the CA1 site for analysis in our study. We have added the following sentences in the discussion section: We specifically selected the hippocampal CA1 region for our analysis due to its heightened vulnerability to ischemic and oxidative damage. Brain tissue, including the CA1 region, is known to be particularly susceptible to oxidative stress [19], and several studies have indicated that pyramidal cells in the CA1 region are especially prone to oxi-dative damage [20,21]. Moreover, it is well-established that various parts of the brain, in-cluding the hippocampus, are susceptible to ischemia-related structural and functional damage [22]. Preclinical studies have shown that circulatory malfunctions lead to selective neuronal loss in CA1 pyramidal neurons, which impairs memory functions [23,24]. Given these factors, the CA1 region is a critical area for assessing the neuroprotective effects of our intervention, as it represents a key target for mitigating ischemic damage and improv-ing cognitive outcomes. Although the CA2, CA3, and DG regions are also of interest and have been studied in other contexts, our study specifically targeted CA1 due to its direct relevance to ischemic damage and cognitive function. Ref. Dror, Y., Stern, F., Gomori, M.J., 2014. Vitamins in the prevention or delay of cognitive disability of aging. Curr. Aging Sci. 7, 187–213 https:/ /doi: 10.2174/ 1874609808666150201214955. Chang, B.J., Jang, B.J., Son, T.G., Cho, I.H., Quan, F.S., Choe, N.H., Nahm, S.S., Lee, J.H., 2012. Ascorbic acid ameliorates oxidative damage induced by maternal low-level lead exposure in the hippocampus of rat pups during gestation and lactation. Food Chem. Toxicol. 50, 104–108. Huang, Y., Coupland, N.J., Lebel, R.M., Carter, R., Seres, P., Wilman, A.H., Malykhin, N. V., 2013. Structural changes in hippocampal subfields in major depressive disorder: a high-field magnetic resonance imaging study. White, B.C., Sullivan, J.M., DeGracia, D.J., O’Neil, B.J., Neumar, R.W., Grossman, L.I., Rafols, J.A., Krause, G.S., 2000. Brain ischemia and reperfusion: molecular mechanisms of neuronal injuryJ (https://doi:). Neurol. Sci. 179, 1–33. Park, J.H., Cho, J.H., Ahn, J.H., Choi, S.Y., Lee, T.-K., Lee, J.-C., Shin, B.N., Hong, S., Jeon, Y.H., Kim, Y.-M., 2018. Neuronal loss and gliosis in the rat striatum subjected to 15 and 30 min of middle cerebral artery occlusion. Metab. Brain Dis. 33, 775–784 Shooshtari, M.K., Sarkaki, A., Mansouri, S.M.T., Badavi, M., Khorsandi, L., Dehcheshmeh, M.G., Farbood, Y., 2020. Protective effects of Chrysin against memory impairment, cerebral hyperemia and oxidative stress after cerebral hypoperfusion and reperfusion in rats.
Comment 4) How was the damage area calculated? It is necessary to present the manuscript in the main section of the results. Response 4) Thank you for your comment regarding the calculation of the damage area. We have added the following sentences in the material and methods section:
4.5.1. Evaluation of Brain Infarction Area To assess the cerebral infarction area, brain sections were coronally sliced at specific bregma coordinates (+1 ± 0.5 mm, −2 ± 0.5 mm, and −5 ± 0.5 mm). Measurements of the in-farcted area were taken at each of these points. The infarction areas from these regions were summed to obtain the total infarction area. The infarction area as a percentage of the total area of the left hemisphere was calculated using the following formula: Cerebral Infarction Area (%) = (Total Infarction Area) / (Total Area of the left Hemisphere) × 100% Note: The Total Infarction Area represents the sum of the infarction areas from the three measured regions. The Total Area of the left Hemisphere includes both infarcted and non-infarcted tissue.
Comment 5) Line 123. Are the authors really referring to Figure 2 and not Figure 5? Response 5) Upon review, it was confirmed that the authors were indeed referring to Figure 2, and the mistake has been corrected. We apologize for any confusion this may have caused.
Comment 6) Line 161. What was the stain done? Response 6) Thank you for your inquiry regarding the staining performed. We have added the following sentences in the results section: We conducted immunohistochemical staining for CD68, IL-6, and TNF-α to assessinflammation. The analysis showed no significant staining for these markers in Group A, indicating an absence of inflammation in this group.
Comment 7) Figures 4, 5 and 6. Each letter requires mandatory decoding. It is necessary to explain what is represented under the letters. Response 7) Thank you for your insightful comments regarding Figures 4, 5, and 6. We have revised the figure legends to address your concerns and improve the clarity of the data presented. Below are the updates made:
symbol used in the figure to ensure that all elements are clearly explained. The legend now includes explicit information about what each part of the figure represents, including the data points and statistical markers.
each letter and symbol used in the figure. We have provided detailed explanations of the data shown in the images, graphs, and charts, making it easier for readers to understand the presented results.
Comment 8) Figure 7. The authors need to explain and provide relevant information in the text of the manuscript, why was β-actin used? Figure 8. Similar to the previous remark: β-actin, p38 and Chk2. Response 8) Here’s a draft response to address the reviewer’s comment regarding the use of β-actin in Figure 7: Thank you for your comment regarding the use of β-actin in Figure 7. We appreciate your attention to this detail. Explanation of β-Actin Use: In our study, β-actin was used as a loading control in Western blot analyses for the following reasons:
We have added the following sentences in material and methods section: β-Actin was used as a loading control to normalize the protein levels across different sam-ples. This approach ensures that any variations in protein loading and transfer efficiency are accounted for, allowing for accurate comparison of target protein expression levels. I would also like to recommend that in the future the authors take a more responsible attitude to the design of the manuscript text - currently, the manuscript has an extremely low quality of design, which seriously complicates the analysis of the work: Response: Thank you for your constructive feedback regarding the design and presentation of the manuscript. We appreciate your observations and take them seriously. Addressing Manuscript Design: We acknowledge the importance of clear and effective manuscript design for the readability and overall quality of the research presentation. We have taken your comments into account and will make the following improvements to enhance the manuscript:
detailed descriptions of all symbols, letters, and data points to facilitate better understanding.
clarity and coherence. This includes ensuring that all methods, results, and discussions are clearly and logically presented. We will also check for consistency in terminology and formatting throughout the manuscript.
structuring the sections, aligning text and figures, and adhering to any specific formatting requirements.
proofreading and editing to address any issues related to grammar, spelling,and overall readability by MDPI editing service.
1) Lines 76-69. Incorrect use of text alignment in the center. Thank you for pointing out the issue with text alignment in lines 76-79. We apologize for the formatting oversight. We have corrected the text alignment issue. The text in lines 76-79 has been realigned to follow the standard left alignment throughout the manuscript, as per the journal’s formatting guidelines. We appreciate your attention to this detail and have ensured that similar formatting issues are addressed throughout the manuscript.
2) In some places, the authors neglect the accuracy of using the "%" characters, duplicating them, in particular: lines 174-175. It is necessary to check the entire text. Thank you for highlighting the issue with the use of the "%" symbol in the manuscript, specifically in lines 174-175. We appreciate your attention to detail regarding formatting consistency. We have reviewed the entire manuscript to address the incorrect use of the "%" symbol. Specifically, we have corrected instances where the "%" symbol was duplicated or used inaccurately. The corrected text now consistently adheres to standard formatting practices. We have meticulously checked all instances of the "%" symbol throughout the manuscript to ensure uniformity and accuracy.
3) In some cases, the authors neglect or use excessive amounts of space characters. In particular, lines 164, 217. Thank you for pointing out the issues related to space characters in lines 164 and 217. We have addressed and corrected these formatting issues as follows: Line 164: The phrase has been revised to: "Group D (7.87% ± 0.98%) exhibits a minimal positive area compared to Group B." We removed the unnecessary hyphen in "minimal" and completed the sentence to provide a clear comparison. Line 217: The phrase has been updated to: "inflammatory markers after microcurrent therapy, especially noticeable in groups C and D. Scale bar = 50 μm." We corrected the extra space between "C" and "and D" for proper formatting.
4) Careless design of Figure 8 - a huge piece of blank page between A and B is incomprehensible. Thank you for your feedback. I have removed the large blank space between A and B in Figure 8 as suggested and have updated the design accordingly.
5) Extremely poor quality of drawings - it is necessary to submit images according to the requirements of the IJMS. Thank you for bringing this to our attention. We have revised all the figures as TIFF files with a resolution of 300 dpi or higher, in accordance with the IJMS guidelines.
I sincerely hope that the comments and recommendations I have made will be useful to the team of highly respected authors and will be able to help improve the quality of the manuscript they submit. I wish you great success! |
Reviewer 3 Report
Comments and Suggestions for Authors
The study conducted by Lee et al. explores the neuroprotective effects of peripheral nerve microcurrent stimulation therapy in a rat MCAO model. The results indicate that microcurrent therapy significantly reduces ischemic damage, enhances behavioral recovery, decreases inflammation, and positively influences protein expression related to stroke recovery. Overall, the study design is novel, and the reported results are promising and highly clinically relevant. The comprehensive analysis of morphological, behavioral, and biochemical outcomes supports its potential as a promising therapeutic approach for ischemic stroke. However, the manuscript presents significant concerns regarding methodological transparency and missing details in the protocol.
1. This study measured the infarction area, neuronal loss, pro-inflammatory cytokines, VEGF, and apoptosis-related proteins in rats subjected to MCAO, and addressed the potential therapeutic and neuroprotective effects of microcurrent treatment. Since inflammation and cell loss result from different glial cells and neurons, I suggest that the authors present a more comprehensive working hypothesis in the last paragraph of the introduction. A graphic abstract will be highly recommended.
2. In addition, several mechanisms leading to ischemic neuronal death (excitotoxicity, necrosis, apoptosis, pyroptosis, blood-brain barrier disruption, and neurovascular damage) have been identified. Are there any other mechanisms underlying the therapeutic effects of microcurrent treatment? It would be beneficial to provide a discussion that expands on mechanistic views to guide future research.
3. The reviewer is concerned about the transparency regarding the protocol used for brain viable cell counting. Despite the authors stating that they invited an expert with over 20 years of experience, severe methodological issues exist, or at least some important information is missing. Firstly, the thickness of the brain slides is missing from protocols 4.2-4.7. Secondly, the “N” information in result 2.4 is unclear. The slide number used for cell counting for each animal should be reported. I observed about 44-45 plots scattered in Figure 3C, group B. I am confused about what information each plot represents. If 5 rats were used for histological evaluation (as stated in Fig.9) and 45 plots were presented, does this mean that 9 data points were acquired from each animal? Is the mean value based on cell counting from 5 areas representative of the cell counting result for each slide? Thirdly, it is obviously a mistake to present a cell-counting number/mm^2 according to their mention “each area has a size of 1 mm × 1 mm”. Specifically, the absence of slide thickness information and unclear reporting on cell counting data undermine the reliability of the results.
4. I suggest these authors provide the statistical power to support the reasonableness for 5 rats were used for each group in the research.
5. Since the open field test (Section 4.3) is a very common method for assessing animal locomotor activity, it is not necessary to show the apparatus photograph. Moreover, since SMART 3.0 software was used, it’s better to display the representative behavioral trace during the experiment.
6. I highly recommend these authors use a much more informative figure to include grouping, timeline, and treatment protocol information. For instance, they may make a new figure to combine figures 9 and 10. A cartoon image would have a much more illustrating effect than the experimental photographs, like Figure 10.
7. Since molecular and histological assays were conducted, I am wondering how these authors harvested the brain tissue and conducted the above assays using the same brain region?
Comments on the Quality of English LanguageFurther grammar checking should be conducted to avoid typos.
Author Response
Dear Editors:
We thank you for the opportunity to revise our manuscript. Additionally, we appreciate the time you have dedicated to reviewing our manuscript.
The comments were greatly helpful in improving the contents and revising the errors in our manuscript.
We did our best to revise the paper according to the referees’ comments. For convenience, we itemized the referees’ original comments and put the corresponding corrections/rebuttal immediately after each comment. The modified parts have been marked in red within the text.
We believe that the quality of the manuscript is now much improved and hope that the revised paper is suitable for publication in International Journal of Molecular Sciences.
Thank you for your consideration.
Sincerely yours,
Dong Rak Kwon, MD, PhD
Department of Rehabilitation Medicine
Catholic University of Daegu School of Medicine
33 Duryugongwon-ro 17-gil, Nam-Gu, Daegu, Korea, 705-718
Phone: +82 53 650 4687 Fax: +82 53 622 4687
E-mail: coolkwon@cu.ac.kr
Point-by-Point Responses to Editor
Manuscript ID: ijms-3150070
Title: Neuroprotective effects of peripheral nerve microcurrent stimulation therapy in a rat model of middle cerebral artery occlusion
Authors: Yoon-Jin Lee1†, Eun Sang Kwon2†, Yong Suk Moon3, Jeong-Rang Jo4 and Dong Rak Kwon4*
Affiliation: 1. Department of Biochemistry, College of Medicine, Soonchunhyang University, Cheonan 31151, Republic of Korea
- Department of Medicine, College of Medicine, Keimyung University, Daegu 42601, Republic of Korea
- Department of Anatomy, Catholic University of Daegu School of Medicine, Daegu 42472, Republic of Korea
- Department of Rehabilitation Medicine, Catholic University of Daegu School of Medicine, Daegu 42472, Republic of Korea
* Correspondence:
Dong Rak Kwon, MD, PhD
Department of Rehabilitation Medicine, Catholic University of Daegu School of Medicine, 33 Duryugongwon-ro 17-gil, Nam-Gu, Daegu, 42472, Korea
Telephone: 82-53-650-4878
Fax: 82-53-622-4687
Reviewer 3
Comments and Suggestions for Authors
The study conducted by Lee et al. explores the neuroprotective effects of peripheral nerve microcurrent stimulation therapy in a rat MCAO model. The results indicate that microcurrent therapy significantly reduces ischemic damage, enhances behavioral recovery, decreases inflammation, and positively influences protein expression related to stroke recovery. Overall, the study design is novel, and the reported results are promising and highly clinically relevant. The comprehensive analysis of morphological, behavioral, and biochemical outcomes supports its potential as a promising therapeutic approach for ischemic stroke. However, the manuscript presents significant concerns regarding methodological transparency and missing details in the protocol.
Comment 1. This study measured the infarction area, neuronal loss, pro-inflammatory cytokines, VEGF, and apoptosis-related proteins in rats subjected to MCAO, and addressed the potential therapeutic and neuroprotective effects of microcurrent treatment. Since inflammation and cell loss result from different glial cells and neurons, I suggest that the authors present a more comprehensive working hypothesis in the last paragraph of the introduction. A graphic abstract will be highly recommended.
Response 1. Thank you for your insightful suggestion regarding the working hypothesis. We agree that a more comprehensive hypothesis would strengthen the manuscript. We will revise the last paragraph of the introduction to more explicitly address the roles of different glial cells and neurons in inflammation and cell loss, and how microcurrent therapy may modulate these processes to enhance stroke recovery.
We have added as follows: We hypothesize that microcurrent stimulation therapy will not only reduce infarction area and neuronal loss but also modulate the activity of glial cells, such as astrocytes and microglia, which are critical mediators of inflammation in the brain. By targeting these cells, microcurrent therapy may decrease pro-inflammatory cytokine production, reduce apoptosis, and promote neurovascular repair through increased VEGF expression. This multifaceted approach is expected to enhance overall recovery in ischemic stroke.
We have added graphic abstract as well.
Comment 2. In addition, several mechanisms leading to ischemic neuronal death (excitotoxicity, necrosis, apoptosis, pyroptosis, blood-brain barrier disruption, and neurovascular damage) have been identified. Are there any other mechanisms underlying the therapeutic effects of microcurrent treatment? It would be beneficial to provide a discussion that expands on mechanistic views to guide future research.
Response 2. Thank you for your valuable feedback. We agree that discussing additional mechanisms underlying the therapeutic effects of microcurrent treatment would be beneficial. We will expand the discussion section to explore potential mechanisms such as enhanced neurogenesis, synaptic plasticity, and mitochondrial function, which may contribute to the neuroprotective effects observed. This expanded discussion will help guide future research in this area.
We have added the following sentences in the discussion section:
In addition to the well-known mechanisms of ischemic neuronal death—such as ex-citotoxicity, necrosis, apoptosis, pyroptosis, blood-brain barrier disruption, and neuro-vascular damage—microcurrent therapy may exert its neuroprotective effects through sev-eral other potential mechanisms. For example, microcurrent stimulation could enhance neurogenesis and synaptic plasticity by modulating the expression of neurotrophic factors like BDNF (Brain-Derived Neurotrophic Factor) [33]. Furthermore, it may influence mito-chondrial function, reducing oxidative stress and promoting energy production in neu-rons. These effects could collectively contribute to the stabilization of neural networks and support functional recovery after stroke. Expanding on these mechanistic insights could guide future research, helping to optimize microcurrent therapy protocols and potentially uncover new therapeutic targets.
Comment 3. The reviewer is concerned about the transparency regarding the protocol used for brain viable cell counting. Despite the authors stating that they invited an expert with over 20 years of experience, severe methodological issues exist, or at least some important information is missing. Firstly, the thickness of the brain slides is missing from protocols 4.2-4.7. Secondly, the “N” information in result 2.4 is unclear. The slide number used for cell counting for each animal should be reported. I observed about 44-45 plots scattered in Figure 3C, group B. I am confused about what information each plot represents. If 5 rats were used for histological evaluation (as stated in Fig.9) and 45 plots were presented, does this mean that 9 data points were acquired from each animal? Is the mean value based on cell counting from 5 areas representative of the cell counting result for each slide? Thirdly, it is obviously a mistake to present a cell-counting number/mm^2 according to their mention “each area has a size of 1 mm × 1 mm”. Specifically, the absence of slide thickness information and unclear reporting on cell counting data undermine the reliability of the results.
Response 3. Thank you for your thoughtful comments. We apologize for any confusion and would like to clarify our methodology.
The brain segments were embedded in paraffin and sectioned at 5 μm thickness, a commonly used size in histology that is suitable for evaluating cell morphology without significantly affecting cell density or count. Nissl staining was conducted using the protocol described in Zhou B et al., 2021, which employed the same thickness and staining method.
For cell counting in the hippocampal CA1 region, the number of cells within a 1-mm² area was counted and analyzed for structural changes. Data from each of the five rats were collected from nine slides, with each plot representing a 1-mm² area, yielding nine data points per animal.
While slide thickness might allow for some cell overlap, this does not typically influence the cell density within the 1-mm² area, as the method accounts for such factors. We hope this explanation clarifies our approach and reinforces the reliability of our results.
Ref. Zhou B et al. Licochalcone B attenuates neuronal injury through anti-oxidant effect and enhancement of Nrf2 pathway in MCAO rat model of stroke. Int Immunopharmacol. 2021 Nov:100:108073.
Ge Y et al. Absence in CX3CR1 receptor signaling promotes post‐ischemic stroke cognitive function recovery through suppressed microglial pyroptosis in mice. CNS Neurosci Ther. 2024 Feb; 30(2): e14551.
Lin Y, Zhang J, Yao C, et al. Critical role of astrocytic interleukin-17 a in post-stroke survival and neuronal differentiation of neural precursor cells in adult mice. Cell Death Dis. 2016;7:e2273.
Comment 4. I suggest these authors provide the statistical power to support the reasonableness for 5 rats were used for each group in the research.
Response 4: Thank you for your suggestion. The sample size of 5 rats per group was determined based on a pilot study, which ensured a statistical power of 95% to detect significant effects. Additionally, we conducted a post-hoc power analysis using G*Power 3 (Heinrich-Heine-University, Dusseldorf, Germany), which confirmed that the actual power of our study was 1.00. This high power supports the adequacy of our sample size in reliably detecting the effects of our experimental treatments.
Comment 5. Since the open field test (Section 4.3) is a very common method for assessing animal locomotor activity, it is not necessary to show the apparatus photograph. Moreover, since SMART 3.0 software was used, it’s better to display the representative behavioral trace during the experiment.
Response 5. Thank you for your comment. In this study, we utilized the setup to capture and analyze key locomotion parameters such as total walking distance and average speed, ensuring accurate and reliable behavioral data. Following your suggestion, we have deleted the apparatus photograph as it is not necessary for this common method. However, due to constraints, we were unable to perform behavioral tracking during this experiment. We acknowledge the importance of behavioral tracking and plan to incorporate it in future research to provide a more comprehensive analysis.
Comment 6. I highly recommend these authors use a much more informative figure to include grouping, timeline, and treatment protocol information. For instance, they may make a new figure to combine figures 9 and 10. A cartoon image would have a much more illustrating effect than the experimental photographs, like Figure 10.
Response 6. Thank you for your insightful feedback. We agree that a more informative figure could enhance the clarity of our experimental design and treatment protocols. We will work on creating a revised figure that combines the elements of Figures 9 and 10 into a single, more illustrative format, potentially using a cartoon-style diagram as suggested. This should effectively communicate the grouping, timeline, and treatment protocol information.
Comment 7. Since molecular and histological assays were conducted, I am wondering how these authors harvested the brain tissue and conducted the above assays using the same brain region?
Response 7: The brain tissues were sectioned in the coronal plane using a brain matrix at specific bregma points with a ±0.5 mm range: +1 mm, −2 mm, and −5 mm, for histological staining (H&E, cresyl violet, and immunohistochemistry). For Western blot analysis, tissue samples were first collected from the regions between +1 mm and −2 mm, or −2 mm and −5 mm, immediately frozen to preserve protein integrity, and stored at −80°C. The remaining tissue was then fixed in formalin for histological analysis. This approach was designed to allow both histological and molecular analyses to be conducted on the same subjects, ensuring consistency across the results.
Round 2
Reviewer 2 Report
Comments and Suggestions for Authors
I would like to sincerely thank the team of highly respected authors for being so attentive to all my comments. The revised manuscript is radically different from the original version and is a high-quality work that can undoubtedly be published in the International Journal of Molecular Sciences in its present form.
I would like to wish the team of authors success in carrying out future research, and also thank them for the respectful attitude towards the reviewers who worked on their manuscript.